# Enhancing patient-centered care: Evaluating quality of life in type 2 diabetes management

Branka Vlahovic[1‡], Vivek Jha[2‡], Vukasin Stojanovic[3], Tanja Vojinovic[4], Arshiya Dutta[5], Pinaki Dutta[2*], Sanja Medenica[6*]

1 Metabolic Intensive Care Unit, Internal Medicine Clinic, Clinical Center of Montenegro, Podgorica, Montenegro, 2 Department of Endocrinology, Post Graduate Institute of Medical Education and Research (PGIMER), Chandigarh, Punjab, India, 3 Emergency Medicine Center of Montenegro, Faculty of Medicine, University of Montenegro, Podgorica, Montenegro, 4 Faculty of Medicine, University of Montenegro, Podgorica, Montenegro, 5 Atal University, Shimla, Himachal Pradesh, India, 6 Department of Endocrinology, Internal Medicine Clinic, Clinical Center of Montenegro, Faculty of Medicine, University of Montenegro, Podgorica, Montenegro

‡ BV and VJ have equally contributed and are joint first author.
* medenicasanja@gmail.com (SM); drpinakidutta12@gmail.com (PD)

## Abstract

### Aims

To evaluate quality of life (QoL) in patients with type 2 diabetes mellitus (T2DM).

### Methods

A cross-sectional study included 151 T2DM patients at the Clinical Centre of Montenegro. The Ferrans and Powers Quality of Life Index (QLI), validated for the Montenegrin population, assessed QoL across five domains. Participants rated items on a Likert scale from 1 (very dissatisfied) to 5 (very satisfied). Data were analysed using SPSS version 22.

### Results

The cohort included 51% women, with a mean age of 60.05 ± 11.63 years. Of the patients, 42% had diabetes for over a decade, and 64% had no additional health conditions. Overall, patients reported satisfactory QoL, especially in self-care and glucose management, though dissatisfaction was high regarding sexual life. Emotional support from family, housing, and friendships significantly contributed to life satisfaction, while financial concerns and job dissatisfaction were common. QoL showed no significant gender differences but declined with age and was notably lower in patients with comorbidities.

### Conclusion

Patients with T2DM report generally satisfactory QoL, with notable concerns in socio-economic and health-related areas. Routine QoL assessments in clinical practice can improve communication, aid in early complication detection, and enable timely interventions to enhance patient outcomes.

**Data availability statement:** All relevant data are within the manuscript and its Supporting Information files.

**Funding:** The author(s) received no specific funding for this work.

**Competing interests:** The authors have declared that no competing interests exist.

## Introduction

Type 2 diabetes mellitus (T2DM) is a widespread chronic disease that has escalated to pandemic proportions. The World Health Organization reports that approximately 422 million people globally are living with diabetes [1]. The International Diabetes Federation projects that by 2045, about 783 million people, or 1 in 8 adults, will have diabetes [2]. T2DM significantly impacts patients' quality of life, leading researchers to investigate health-related quality of life (HRQOL) and its influencing factors [3]. Regular assessments of quality of life in clinical practice can enhance communication between patients and healthcare providers, uncover hidden issues, and assess the impact of treatments on individual patients [4].

Managing T2DM requires daily commitment to lifestyle changes, including a healthy diet, physical activity, smoking cessation, regular check-ups, blood glucose monitoring, and medication adherence if prescribed. This constant self-management can restrict patients' lives, affecting their emotional and social well-being as well as their financial stability due to medication and frequent doctor visits, which can be costly and disruptive [5]. Each individual with diabetes faces unique challenges, and while many struggle with disease control, all experience significant life impacts. Patients often feel mentally burdened by the strict regimen their condition imposes. Additional challenges include microvascular and macrovascular complications associated with short- and long-term diabetes management [6].

Lower quality of life among diabetes patients is linked to factors such as older age, female gender, low economic status, limited education, inadequate physical activity, and chronic complications from the disease [7]. Effective management of diabetes involves a holistic approach that balances mental health, family support, and blood glucose control [8]. Current trends highlight the importance of primary prevention as a cost-effective strategy. Prevention efforts should concentrate on urban planning, food marketing policies, and fostering supportive environments in workplaces and schools to promote healthier lifestyles. Targeting these interventions towards children and families could significantly reduce the risk of T2DM in future generations [9].

This study aims to evaluate how T2DM affects the quality of life (QoL) of patients using the adapted Ferrans and Powers Quality of Life Index (QLI). Specifically, we seek to determine which aspects of QoL are most impacted by diabetes management and to assess the effectiveness of this tool in capturing the comprehensive effects of the disease on patients' daily lives and well-being

## Subjects and methods

The study was conducted as a cross-sectional analysis from October 23, 2023, to December 23, 2023, at the Internal Clinic and Polyclinic of the Clinical Center of Montenegro. It involved 151 patients diagnosed with type 2 diabetes mellitus (T2DM) receiving treatment at the same facility. Prior to participation, all subjects were thoroughly informed about the study's objectives and procedures. They voluntarily consented to participate with written consent, understanding their right to withdraw at any time. To ensure confidentiality, the survey was conducted anonymously under conditions that safeguarded participants' privacy.

### Ethical approval

The study was conducted in accordance with the ethical standards of the Clinical Center of Montenegro and the Declaration of Helsinki, with approval from the institution's Ethics Committee, decision number EC:03/01-9732/1. No notable ethical concerns were identified during the research process.

## Clinical methodology

To assess the quality of life of patients with type 2 diabetes mellitus (T2DM), we utilized a modified and validated version of the Ferrans and Powers Quality of Life Index (QLI) tailored for the Montenegrin population. The QLI measures life satisfaction across various domains, weighted by the importance individuals assign to each aspect. Our questionnaire comprised five sections with a total of 50 questions. Participants responded using a Likert scale ranging from 1 (very dissatisfied/unimportant) to 5 (very satisfied/important). This approach provided a nuanced understanding of how different life domains impact overall quality of life from the individual's perspective.

The five sections of the questionnaire included:

1. **Demographic Information**: Collected participants' demographic details.

2. **Life Satisfaction**: Assessed participants' overall satisfaction with life.

3. **Environmental and Daily Activity Satisfaction**: Evaluated contentment with surroundings and daily routines.

4. **Importance of Daily Activities**: Measured the significance participants place on their daily activities.

5. **Importance of Environment and Daily Activities**: Explored the perceived importance of the environment and daily activities.

In addition to the modified Ferrans and Powers Quality of Life Index (QLI), several other assessment tools are widely used:

A. **World Health Organization Quality of Life Instruments (WHOQOL)**: Developed by the WHO, these tools are designed for cross-cultural applicability and have been utilized in various populations, including those in Montenegro.

B. **EQ-5D**: A standardized measure of health-related quality of life developed by the EuroQol Group, EQ-5D has been widely used in clinical and economic appraisals across different populations.

C. **SF-36 Health Survey**: A 36-item, patient-reported survey assessing eight domains of health, commonly used in clinical practice and research to evaluate overall health status.

D. **Patient-Reported Outcomes Measurement Information System (PROMIS)**: Provides measures of physical, mental, and social health across various chronic conditions and the general population.

E. **Nottingham Health Profile (NHP)**: A general patient-reported outcome measure designed to assess perceived health status across multiple domains.

The Ferrans and Powers Quality of Life Index (QLI) provides a personalized assessment by weighting satisfaction across various life domains health, psychological well-being, social/economic factors, and family, according to individual importance. This approach ensures that the evaluation reflects personal values and priorities, offering a comprehensive understanding of quality of life. In contrast, instruments like the WHOQOL and EQ-5D, while valuable, may not fully capture individual priorities. The QLI's strong construct validity and high internal consistency reliability make it particularly suitable for diverse populations [10].

We selected the Ferrans and Powers Quality of Life Index (QLI) for its ability to be customized for specific populations, making it highly relevant for assessing the impacts of Type 2 diabetes mellitus (T2DM) on quality of life. For our Montenegrin cohort, we adapted the QLI

to focus on aspects crucial to diabetes management, such as dietary restrictions, medication adherence, and the psychological impacts of living with a chronic condition. This adaptation involved modifying the QLI based on consultations with local healthcare professionals and feedback from a pilot group of the study population, ensuring the tool's cultural and contextual relevance. These adjustments allow the QLI to accurately reflect the diabetes-related quality of life challenges specific to Montenegro, enhancing the instrument's sensitivity and the validity of our data.

## Study population

The study comprised 151 patients with type 2 diabetes mellitus (T2DM) receiving treatment at the Internal Clinic and Polyclinic of the Clinical Center of Montenegro.

## Statistical analyses

Data analysis was meticulously carried out using SPSS software, version 22. The statistical techniques employed were chosen to appropriately address the nature of the data and the research questions posed. Initially, descriptive statistics were computed to provide an overview of the demographic characteristics and key variables. This included the calculation of means and standard deviations for continuous variables, and frequencies and percentages for categorical variables.

For inferential analysis, the Mann-Whitney U test was utilized to compare differences in quality of life scores between two independent groups, particularly focusing on the presence or absence of comorbidities. Additionally, the Kruskal-Wallis test was applied to explore differences among multiple independent groups, such as variations in quality of life across different durations of diabetes.

To investigate associations between continuous variables, Pearson's correlation coefficient (r) was calculated. This provided insights into the relationships between age, duration of diabetes, and quality of life scores. A significance level of $p < 0.05$ was set for all statistical tests to determine the presence of statistically significant findings.

The results of these analyses were visually represented through both tabular and graphical formats to facilitate a clear understanding of the data trends and statistical relationships.

## Results

The initial section of the questionnaire focused on the demographic characteristics of the participants (refer to Supplementary file; S1 Table). Among the 151 individuals surveyed, 74 (49%) were male. The average age was 60.05 ± 11.63 years, ranging from 34 to 84 years. As for educational attainment, 45.7% had completed secondary school, 14.6% had only primary education, 12.6% held higher education qualifications, 11.3% possessed university degrees, and 4.0% had not completed elementary school. Regarding marital status, 54.7% were married, 16.7% were widowed, 13.3% lived alone, and 12.0% were divorced. Concerning the duration of diabetes, 42% had been diagnosed for over 10 years, 34% for less than 5 years, and 24% for between 5 and 10 years. Additionally, 64% reported no other diseases, while 36% indicated the presence of comorbidities.

The second section of the study assessed participants' subjective life satisfaction across various domains of entire study population, using a scale from 1 (very dissatisfied) to 5 (very satisfied). Key findings include:

- **Personal Health Satisfaction**: 34.4% of participants were satisfied, while 20.5% expressed dissatisfaction, resulting in an average score of 3.14 ± 0.98.

- **Healthcare Satisfaction**: A majority of 51.3% were satisfied, with an average score of 3.56 ± 0.79.

- **Energy Levels for Daily Activities**: 33.8% reported satisfaction, whereas 24.5% were dissatisfied, leading to an average score of 3.28 ± 1.02.

- **Self-Care Ability Without Assistance**: 47.7% were satisfied, and 20.5% were very satisfied, averaging a score of 3.70 ± 1.03.

- **Independent Glucose Monitoring at Home**: 55.0% expressed satisfaction, with an average score of 3.81 ± 0.91.

- **Satisfaction with Lifestyle Changes Due to Diabetes**: 40.4% were satisfied, while 14.6% were dissatisfied, resulting in an average score of 3.52 ± 0.95.

- **Control Over Personal Health**: 49.7% felt satisfied, leading to an average score of 3.57 ± 0.96.

- **Sexual Life Satisfaction**: 32.6% were satisfied, whereas 17.4% were dissatisfied, with an average score of 3.04 ± 1.17.

Detailed findings for each domain are presented in Table 1.

In the third domain of the questionnaire, respondents assessed their satisfaction with various aspects of their environment and daily activities. Notably, 82.4% of respondents expressed satisfaction with the emotional support received from family, with an average score of 4.07 ± 0.89. Similarly, 83.1% were satisfied with their living arrangements, resulting in an average score of 3.99 ± 0.78. In contrast, satisfaction with spouses or partners was comparatively lower, with only 60.5% reporting contentment and an average score of 3.04 ± 1.17. Overall, the mean satisfaction score across all evaluated aspects was 3.68 ± 0.72, indicating a trend toward neutrality or slight satisfaction among respondents. Detailed percentages and mean scores for each aspect are presented in Table 2.

The Mann-Whitney U test demonstrated a significant effect of comorbidities on quality of life, with a Z value of 2.84 and a p-value of less than 0.001. Participants without comorbidities reported higher satisfaction (Median = 85.52) compared to those with comorbidities (Median = 75.20). Additionally, no significant associations were found between quality of life and either gender or the duration of the disease (refer to Supplementary file; S2 Table).

Pearson's linear correlation coefficient was calculated to assess the relationship between age and quality of life among the respondents, revealing a robust negative correlation, r=−0.263, with a p-value of less than 0.001. This indicates that satisfaction with quality of life decreases as age increases (refer to Supplementary file; S3 Table).

**Table 1. Subjective satisfaction levels of the study participants.**

| Aspect of Satisfaction | Very Dissatisfied (%) | Dissatisfied (%) | Neutral (%) | Satisfied (%) | Very Satisfied (%) | Mean ± SD |
|---|---|---|---|---|---|---|
| Personal Health | 5.5 | 20.5 | 34.4 | 34.4 | 5.3 | 3.14 ± 0.98 |
| Healthcare Services | 2.0 | 6 | 33.3 | 51.3 | 7.3 | 3.56 ± 0.79 |
| Energy for Daily Activities | 2.0 | 24.5 | 28.5 | 33.8 | 11.3 | 3.28 ± 1.02 |
| Ability to Self-Care Without Assistance | 0.0 | 11.9 | 16.6 | 47.7 | 20.5 | 3.70 ± 1.03 |
| Ability to Independently Monitor Glucose Levels at Home | 3.3 | 5.3 | 17.2 | 55.0 | 19.2 | 3.81 ± 0.91 |
| Adjustments Made Due to Diabetes (e.g., diet, exercise) | 1.3 | 14.6 | 29.1 | 40.4 | 14.6 | 3.52 ± 0.95 |
| Control Over Personal Health | 4.8 | 6.9 | 26.9 | 49.7 | 11.7 | 3.57 ± 0.96 |
| Sexual Life | 13.8 | 17.4 | 28.3 | 32.6 | 8.0 | 3.04 ± 1.17 |

**Table 2. Satisfaction with environment and daily activities.**

| Aspect Evaluated | No of Responders (N) | Very Dissatisfied (%) | Dissatisfied (%) | Neutral (%) | Satisfied (%) | Very Satisfied (%) | Mean Score ± SD |
|---|---|---|---|---|---|---|---|
| Spouse/Partner Relationship | 139 | 7.9 | 15.8 | 15.8 | 43.2 | 17.3 | 3.04 ± 1.17 |
| Emotional Support from Family | 148 | 2.7 | 2.7 | 12.2 | 49.3 | 33.1 | 4.07 ± 0.89 |
| Friendships | 148 | 2.7 | 4.0 | 22.1 | 51.7 | 19.5 | 3.81 ± 0.88 |
| Emotional Support from Non-Family | 148 | 3.3 | 10.6 | 31.8 | 44.4 | 9.9 | 3.47 ± 0.93 |
| Ability to Fulfill Family Obligations | 146 | 2.7 | 6.2 | 19.9 | 54.8 | 9.9 | 3.76 ± 0.89 |
| Perceived Usefulness to Others | 150 | 4.0 | 6.7 | 18.7 | 54.7 | 16.4 | 3.72 ± 0.94 |
| Satisfaction with Living Situation | 148 | 2.0 | 2.0 | 12.8 | 61.5 | 21.6 | 3.99 ± 0.78 |
| Job Satisfaction | 111 | 5.4 | 12.6 | 22.5 | 46.8 | 12.6 | 3.49 ± 1.04 |
| Ability to Meet Financial Needs | 150 | 5.4 | 8.1 | 28.9 | 47.0 | 10.7 | 3.44 ± 0.98 |
| Engagement in Leisure Activities | 149 | 2.7 | 2.7 | 19.3 | 59.3 | 21.6 | 3.50 ± 0.97 |
| Peace of Mind | 150 | 2.8 | 6.2 | 27.6 | 52.4 | 11.0 | 3.83 ± 0.83 |
| Achievement of Personal Goals | 145 | 2.7 | 0.7 | 24.8 | 52.3 | 19.5 | 3.63 ± 0.86 |
| Overall Happiness | 149 | 2.0 | 9.9 | 31.1 | 43.7 | 13.2 | 3.83 ± 0.83 |
| Physical Appearance | 151 | 2.7 | 0.7 | 24.8 | 52.3 | 19.5 | 3.56 ± 0.91 |

The fourth section of our questionnaire evaluated respondents' perceptions of various aspects of their daily lives. Table 3 summarizes the average ratings for the entire study population across various categories, including personal health, healthcare, energy levels for daily activities, independent self-care, self-monitoring of blood glucose, lifestyle changes due to diabetes, sexual life, the role of a spouse or partner, emotional support from family, the importance of friends, and emotional support from non-family members. The data reveal that respondents prioritize personal health (Mean = 4.32, SD = 0.76) and emotional support from family (Mean = 4.20, SD = 0.84) the highest. Conversely, aspects such as sexual life (Mean = 3.27, SD = 1.27) and emotional support from non-family members (Mean = 3.58, SD = 0.94) are deemed less important.

The fifth section of our questionnaire assessed respondents' perceptions of the importance of various aspects of their environment and daily activities. Table 4 summarizes the average ratings for the following elements: being useful to others, absence of worries, neighborhood, home or residence, job, ability to manage financial needs, engaging in enjoyable activities, peace of mind, achievement of personal goals, physical appearance, and overall happiness.

**Table 3. Respondents' subjective perceptions of the importance of various aspects of their daily lives.**

| Aspect Evaluated | Mean | SD |
|---|---|---|
| Own health | 4.32 | 0.76 |
| Healthcare | 4.23 | 0.79 |
| Energy for daily activities | 4.05 | 0.81 |
| Ability to self-care without assistance | 4.20 | 0.75 |
| Ability to independently monitor blood glucose | 4.01 | 0.80 |
| Lifestyle changes due to diabetes | 3.92 | 0.78 |
| Sexual life | 3.27 | 1.27 |
| Spouse or partner | 3.72 | 1.27 |
| Emotional support from family | 4.20 | 0.84 |
| Friends | 3.90 | 0.81 |
| Emotional support from non-family members | 3.58 | 0.94 |

**Table 4. Respondents' perceived importance of environmental factors and daily activities.**

| Aspect | Mean | SD |
|---|---|---|
| Being useful to others | 3.99 | 0.72 |
| Absence of worries | 4.04 | 0.75 |
| Neighborhood | 3.34 | 0.70 |
| Home, apartment or place of residence | 4.10 | 0.70 |
| Job | 3.79 | 1.04 |
| Ability to manage financial needs | 3.99 | 0.82 |
| Engaging in enjoyable activities | 3.68 | 0.91 |
| Peace of mind | 4.22 | 0.79 |
| Achieving personal goals | 4.09 | 0.73 |
| Physical appearance | 3.71 | 0.92 |
| Overall happiness | 4.30 | 0.75 |

The analysis of the entire study population reveals that respondents place the highest importance on overall happiness (Mean = 4.30, SD = 0.75) and peace of mind (Mean = 4.22, SD = 0.79). In contrast, aspects such as satisfaction with their neighborhood (Mean = 3.34, SD = 0.70) and engagement in enjoyable activities (Mean = 3.68, SD = 0.91) are considered less significant.

## Discussion

Diabetes is a chronic illness, necessitating regular assessments of patients' quality of life. Complications from diabetes impact multiple organ systems, contributing significantly to the disease's associated morbidity and mortality [11]. Quality of life is typically diminished among individuals with diabetes, irrespective of gender. Studies on quality-of-life aid in evaluating psychological functioning, identifying patients' specific needs and deficits at different stages of T2DM, and comparing the effects of various treatment regimens on well-being and satisfaction [12]. In our study, most respondents were female (51%), with an average age of 60.05 ± 11.63 years, a finding consistent with Ali et al., who reported a mean age of 59.65 ± 12.3 years in their research [13]. Wild et al. noted that, in developing countries, the highest prevalence of diabetes is among individuals aged 45 to 65 years [14]. Younger patients, despite high glucose and HbA1c levels, tend to experience a better quality of life, likely due to a shorter disease duration and fewer complications. This age group also benefits from better healthcare access and stronger family support, which fosters a more optimistic outlook. Several studies have observed that young individuals with diabetes enjoy a quality of life comparable to their non-diabetic peers [15]. Our findings indicate that 64% of respondents do not have other illnesses, while over one-third report having at least one comorbidity.

These findings align with studies indicating an aging population, which correlates with increased rates of chronic illness and higher medication usage [16]. The reduced quality of life observed in individuals with diabetes is partly due to the fact that elderly patients often have multiple chronic conditions, leading to polypharmacy and an elevated risk of cognitive complications [17].

In our study, quality of life was assessed in two domains, each consisting of 11 questions. The first domain focused on satisfaction with health, while the second assessed satisfaction with environment and socio-economic conditions. In the health and functioning domain, respondents expressed the greatest dissatisfaction with their sexual life, which aligns with the known impact of diabetes on sexual function. Hyperglycemia, a key factor in diabetic

macrovascular and microvascular complications, contributes to the mechanisms of sexual dysfunction. Additionally, diabetes-related conditions such as hypertension, dyslipidemia, obesity, metabolic syndrome, and smoking are risk factors for sexual dysfunction in both genders [18].

Respondents reported the highest satisfaction with their ability to care for themselves independently and manage their blood glucose levels. In contrast, they expressed the most dissatisfaction in the socio-economic domain, particularly with their ability to meet financial needs. One study highlights the broader impact of diabetes on patients, employers, and society, noting not only higher unemployment rates but also health-related work limitations for those who remain employed [19]. The greatest sources of satisfaction were emotional support from family, their living environment, and friendships, reflecting a positive assessment of their social domain. The psychological outlook of individuals with diabetes plays a crucial role in their engagement with disease management [20]. These results underscore the importance of family support, aligning with previous findings that strong family relationships positively influence metabolic control [21].

Analyzing the average scores of the two questionnaire domains, respondents indicated slightly higher satisfaction with their environment and socio-economic conditions (3.68 ± 0.72) than with their health (3.45 ± 0.73). The overall quality of life was rated at an average of 3.65, suggesting that most respondents are generally satisfied. However, as most participants are elderly, the lower scores in quality of life could be attributed to age-related challenges and the prevalence of diabetes with comorbidities in older populations. Similar findings were noted by Ali et al. and Glasgow et al., who observed that aging negatively impacts quality of life in diabetes patients; our study also found a weak negative correlation between age and quality of life (p = 0.00) [13,22], indicating that satisfaction tends to decrease with age. For accurate assessments, the effect of age must be considered when analyzing diabetes and quality of life. Additionally, our research found no significant differences in quality of life based on gender, nor was the duration of diabetes significantly linked to quality of life, consistent with findings in other studies [23,24].

Our study enhances the current research on the quality of life (QoL) in individuals with Type 2 diabetes mellitus (T2DM) by utilizing the adapted Ferrans and Powers Quality of Life Index (QLI). This adaptation allows us to explore a broad array of life domains impacted by diabetes management, such as personal health and emotional support, affirming many earlier findings about diabetes' impact on QoL.

In contrast to other QoL instruments used in diabetes research like the World Health Organization Quality of Life (WHOQOL) instruments, EQ-5D, SF-36 Health Survey, and the Diabetes Quality of Life (DQOL) measure, the QLI offers unique insights. For example, WHOQOL facilitates broad cultural comparisons, EQ-5D supports economic evaluations, and SF-36 provides detailed health status assessments. Meanwhile, the DQOL, tailored specifically for diabetes, focuses on disease-specific lifestyle impacts and treatment satisfaction.

Our results particularly align with those from the DQOL, highlighting significant areas like self-care and glucose management as crucial for life satisfaction. However, the QLI's integration of personal value weightings adds a nuanced layer to the QoL assessment, emphasizing patient-centered factors such as emotional and familial support. This is a dimension often corroborated by studies using the SF-36, which notes the importance of social functioning and emotional roles.

Looking forward, conducting cross-instrument comparisons could offer deeper insights into how various cultural and healthcare contexts influence the prioritization of different QoL dimensions, potentially broadening our understanding of effective diabetes management worldwide.

To gain a more comprehensive understanding of the quality of life in patients with T2DM, future research should aim to compare these findings with those from individuals without diabetes.

These insights underscore the importance of addressing both medical and socio-economic factors in diabetes management, particularly as the aging population grows. Future research should include comparisons with non-diabetic populations to gain a broader understanding of how T2DM uniquely impacts quality of life.

## Conclusion

In conclusion, this study reveals that the overall quality of life in individuals with type 2 diabetes mellitus (T2DM) is generally satisfactory, with notable variations across different life domains. In the health-related domain, participants reported the highest satisfaction with their independence in self-care and blood glucose management, while expressing the most dissatisfaction with their sexual life. In the socio-economic domain, emotional support from family, living environment, and friendships were key sources of satisfaction. However, financial difficulties and job dissatisfaction were prominent areas of concern.

Our findings show that quality of life in T2DM patients declines with age, indicating that older individuals experience greater challenges, likely due to age-related health issues and comorbidities. Importantly, the study found no significant difference in quality of life based on gender or diabetes duration. However, the presence of comorbidities significantly impacted quality of life, with individuals without additional health conditions reporting higher satisfaction.

## Supporting information

**S1 Table. Demographic characteristics of participants.**
(DOCX)

**S2 Table. Association between quality of life in T2DM patients and variables: gender, disease duration, and comorbidity.** (*Kruskal-Wallis test; **Mann-Whitney U test).
(DOCX)

**S3 Table. The influence of age on the quality of life of respondents.**
(DOCX)

**S1 Questionnaire. Inclusivity in global research questionnaire final.**
(DOCX)

**S1 Checklist. PLOS One human subjects research checklist.**
(DOCX)

## Author contributions

**Conceptualization:** Branka Vlahovic, Sanja Medenica.

**Data curation:** Vivek Jha.

**Formal analysis:** Branka Vlahovic, Vivek Jha.

**Investigation:** Branka Vlahovic, Vukasin Stojanovic, Tanja Vojinovic.

**Methodology:** Branka Vlahovic, Vivek Jha, Vukasin Stojanovic, Tanja Vojinovic, Arshiya Dutta, Pinaki Dutta, Sanja Medenica.

**Resources:** Tanja Vojinovic, Sanja Medenica.

**Supervision:** Branka Vlahovic, Vukasin Stojanovic, Tanja Vojinovic, Pinaki Dutta, Sanja Medenica.

**Validation:** Branka Vlahovic, Vivek Jha, Vukasin Stojanovic, Tanja Vojinovic, Arshiya Dutta, Pinaki Dutta, Sanja Medenica.

**Visualization:** Branka Vlahovic, Vukasin Stojanovic, Pinaki Dutta, Sanja Medenica.

**Writing – original draft:** Vivek Jha, Arshiya Dutta.

**Writing – review & editing:** Branka Vlahovic, Vivek Jha, Vukasin Stojanovic, Tanja Vojinovic, Arshiya Dutta, Pinaki Dutta, Sanja Medenica.

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
