## [Decision Letter · Decision Letter 0]

2 Jan 2025

PONE-D-24-59232Enhancing Patient-Centered Care: Evaluating Quality of Life in Type 2 Diabetes ManagementPLOS ONE

Dear Dr. Dutta,

Thank you for submitting your manuscript to PLOS ONE. After careful consideration, we feel that it has merit but does not fully meet PLOS ONE’s publication criteria as it currently stands. Therefore, we invite you to submit a revised version of the manuscript that addresses the points raised during the review process.

We look forward to receiving your revised manuscript.

Kind regards,

Hidetaka Hamasaki

Academic Editor

PLOS ONE

Reviewers' comments:

Reviewer's Responses to Questions

**Comments to the Author**

1. Is the manuscript technically sound, and do the data support the conclusions?

Reviewer #1: Yes

Reviewer #2: Yes

2. Has the statistical analysis been performed appropriately and rigorously? 

Reviewer #1: Yes

Reviewer #2: Yes

3. Have the authors made all data underlying the findings in their manuscript fully available?

Reviewer #1: Yes

Reviewer #2: Yes

4. Is the manuscript presented in an intelligible fashion and written in standard English?

Reviewer #1: Yes

Reviewer #2: Yes

5. Review Comments to the Author

Reviewer #1: Introduction

Relevance: The Introduction effectively sets the stage by outlining the importance of assessing quality of life (QoL) in managing chronic conditions such as type 2 diabetes mellitus (T2DM). It successfully ties the concept of QoL to patient-centered care, which is pertinent to the manuscript's goals.

Clarity and Language Quality: The English is generally clear. However, some sentences could be more concise, and the flow could be improved by reducing redundancy and better organizing the information. Specific attention to transitions between topics would enhance readability.

Methods

Relevance: The Methods section is relevant, detailing the cross-sectional study design, the population studied, and the instruments used for data collection. However, the choice of the QLI and its adaptation for the Montenegrin population requires further justification, particularly regarding its specificity for diabetes-related QoL issues.

Clarity and Language Quality: This section is straightforward but would benefit from more detailed descriptions of the data analysis procedures. The English is functional but could be enhanced by refining technical descriptions and using more precise statistical language to improve the professional tone.

Results

Relevance: The Results section appropriately presents the data on QoL across various domains. The inclusion of specific data on areas like self-care, glucose management, and emotional support is highly relevant and aligns well with the study’s objectives.

Clarity and Language Quality: The presentation of results is clear, with effective use of tables and figures to summarize findings. The language is mostly clear, though some parts could benefit from more careful proofreading to correct minor grammatical errors and ensure consistency in terminology.

Discussion

Relevance: The Discussion makes relevant connections between the findings and broader QoL research. However, it could be improved by including a more comprehensive comparison with existing literature, especially concerning different QoL instruments used in diabetes research.

Clarity and Language Quality: The Discussion is insightful but occasionally drifts into generalization. More precise language could help in directly linking the study’s results to implications for clinical practice. Enhancements in word choice and sentence structure would also elevate the overall professional quality of the manuscript.

Overall Language Quality

The manuscript's language quality is adequate for scholarly communication but requires some improvements to meet high academic standards. There are occasional grammatical mistakes and awkward phrasings that could distract from the content. A thorough proofreading by a native English speaker or a professional editor would be beneficial to polish the text and ensure it conveys the intended meaning clearly and professionally.

Conclusion

The manuscript is fundamentally sound but could be significantly improved by addressing the specific issues noted in each section. Recommendations for revisions include strengthening the justification for the methodology, enhancing the analytical depth of the discussion, and improving the linguistic quality throughout the document. With these changes, the manuscript would be a stronger candidate for publication, offering valuable insights into the QoL of patients with T2DM.

Reviewer #2: The authors evaluated the quality of life (QoL) across five domains in Montenegrin patients with type 2 diabetes mellitus (T2DM) by using the validated Ferrans and Powers Quality of Life Index (QLI). The paper is very well-written and important for clinical practice. An Intro is very informative, the Results section is correct, the Discussion is brilliant and the conclusions with future perspectives arise from the data presented. The references are correct and updated. After some minor, cosmetic changes suggested by the reviewer, it is absolutely suitable for publishing in PlosOne.

Issues to be addressed:

1. Intro: please, add one to two sentences that cover the aim of presented paper.

2. Methods: Please, add institutional EC decision #

- please, explain what the adaptation of questionnaire means?

- Pearson's correlation coefficient is r. Please, correct.

3. Results: I mean that in the Tables 1, 3, 4, the column No of respondents (n=151) should be deleted after it is emphasized and explained in the text linked to the tables (i.e. entire study population are the respondents).

- What the letter H means in Mann-Whitney test (it can be U or Z)(the text below the Table 2)? Please, correct or explain if it is possible.

- Add the Pearson's coefficient r in front of -263 in the form: r=-0.263, p<0.00. In presented form, such correlation is not statistically weak, as you mentioned. On contrary, it is a robust negative correlation. Please, correct!

6. PLOS authors have the option to publish the peer review history of their article (what does this mean? ). If published, this will include your full peer review and any attached files.

**Do you want your identity to be public for this peer review?** For information about this choice, including consent withdrawal, please see our Privacy Policy .

Reviewer #1: **Yes: ** Izabella Uchmanowicz

Reviewer #2: No

---

## [Author Response · Author response to Decision Letter 0]

7 Jan 2025

Reviewer 1 Comments and Responses

Introduction

Comment: The English is generally clear. However, some sentences could be more concise, and the flow could be improved by reducing redundancy and better organizing the information. Specific attention to transitions between topics would enhance readability.

Response: As suggested, the Introduction has been thoroughly revised to improve conciseness and flow.

Changes: Page 3, Lines 94-101, 111-119.

Methods

Comment: The Methods section is relevant, but the choice of the QLI and its adaptation for the Montenegrin population requires further justification.

Response: Justification for the choice of QLI and its adaptation is now included, emphasizing its relevance and the modifications made for the Montenegrin context.

Changes: Page 5, Lines 218-228.

Methods - Clarity and Language Quality

Comment: This section would benefit from more detailed descriptions of the data analysis procedures.

Response: Additional details about data analysis have been incorporated to enhance clarity.

Changes: Page 6, Lines 238-252.

Results - Language Quality

Comment: Some parts could benefit from more careful proofreading to correct minor grammatical errors and ensure consistency in terminology.

Response: Language in the results section has been corrected as advised.

Changes: Page 6, Lines 254-262; Page 8, Lines 319-325; Page 9, Lines 355-360.

Discussion

Comment: The Discussion should include a more comprehensive comparison with existing literature, especially concerning different QoL instruments used in diabetes research.

Response: A more comprehensive comparison with existing literature has been implemented as suggested.

Changes: Page 12, Lines 483-495.

Discussion - Clarity and Language Quality

Comment: More precise language could help in directly linking the study’s results to implications for clinical practice.

Response: Proofreading has been done as suggested to enhance clarity and professional quality.

Changes: Page 12-13, Lines 483-501.

Reviewer 2 Comments and Responses

Introduction - Aim

Comment: Please add one to two sentences that cover the aim of the presented paper.

Response: Added 1-2 sentences to the introduction to clearly state the aim.

Changes: Page 3, Lines 120-123.

Methods - EC Decision

Comment: Please add institutional EC decision number.

Response: EC decision number has been included.

Changes: Page 4, Line 168.

Methods - Explanation of Adaptation

Comment: Please explain what the adaptation of the questionnaire means.

Response: Explanation of questionnaire adaptation has been detailed.

Changes: Page 4, Line 177.

Pearson's Correlation Coefficient

Comment: Pearson's correlation coefficient should be corrected.

Response: Corrected as suggested.

Changes: Page 6, Line 247.

Results - Tables

Comment: The column 'No of respondents (n=151)' should be deleted from Tables 1, 3, 4 after it is emphasized in the text linked to the tables.

Response: Column removed from tables and explained in the linked text.

Changes: Pages 7, 10, 11.

Statistical Notation Correction

Comment: What does the letter H mean in the Mann-Whitney test?

Response: Mistake corrected; it should have been Z.

Changes: Page 9, Line 345.

Statistical Correlation

Comment: Add the Pearson's coefficient r in front of -263 in the form: r = -0.263, p < 0.00.

Response: Corrected to reflect the robust negative correlation.

Changes: Page 9, Lines 350-353.

---

## [Decision Letter · Decision Letter 1]

2 Feb 2025

Enhancing patient-centered care: evaluating quality of life in type 2 Diabetes management

PONE-D-24-59232R1

Dear Dr. Dutta,

We’re pleased to inform you that your manuscript has been judged scientifically suitable for publication and will be formally accepted for publication once it meets all outstanding technical requirements.

Kind regards,

Hidetaka Hamasaki

Academic Editor

PLOS ONE

Additional Editor Comments (optional):

Reviewers' comments:

Reviewer's Responses to Questions

**Comments to the Author**

1. If the authors have adequately addressed your comments raised in a previous round of review and you feel that this manuscript is now acceptable for publication, you may indicate that here to bypass the “Comments to the Author” section, enter your conflict of interest statement in the “Confidential to Editor” section, and submit your "Accept" recommendation.

Reviewer #2: All comments have been addressed

2. Is the manuscript technically sound, and do the data support the conclusions?

Reviewer #2: Yes

3. Has the statistical analysis been performed appropriately and rigorously? 

Reviewer #2: Yes

4. Have the authors made all data underlying the findings in their manuscript fully available?

Reviewer #2: Yes

5. Is the manuscript presented in an intelligible fashion and written in standard English?

Reviewer #2: Yes

6. Review Comments to the Author

Reviewer #2: The quality of the papers was substantially improved by the authors' thorough response to the reviewers' comments. The English is accurate.

Yes to me.

7. PLOS authors have the option to publish the peer review history of their article (what does this mean? ). If published, this will include your full peer review and any attached files.

**Do you want your identity to be public for this peer review?** For information about this choice, including consent withdrawal, please see our Privacy Policy .

Reviewer #2: No

---

## [Editor Report · Acceptance letter]

PONE-D-24-59232R1

PLOS ONE

Dear Dr. Dutta,

I'm pleased to inform you that your manuscript has been deemed suitable for publication in PLOS ONE. Congratulations! Your manuscript is now being handed over to our production team.

Kind regards,

on behalf of

Dr. Hidetaka Hamasaki

Academic Editor

PLOS ONE